# The Value of Urinary NGAL, KIM-1, and IL-18 Measurements in the Early Detection of Kidney Injury in Oncologic Patients Treated with Cisplatin-Based Chemotherapy

**DOI:** 10.3390/ijms25021074

**Published:** 2024-01-16

**Authors:** Dawid Szumilas, Aleksander Jerzy Owczarek, Aniceta Brzozowska, Zofia Irena Niemir, Magdalena Olszanecka-Glinianowicz, Jerzy Chudek

**Affiliations:** 1Department of Internal Diseases and Oncological Chemotherapy, Faculty of Medicine in Katowice, Medical University of Silesia in Katowice, 40-027 Katowice, Poland; chj@poczta.fm; 2Health Promotion and Obesity Management Unit, Department of Pathophysiology, Faculty of Medicine in Katowice, Medical University of Silesia in Katowice, 40-055 Katowice, Poland; aowczarek@sum.edu.pl (A.J.O.); anicetabrz@wp.pl (A.B.); molszanecka@sum.edu.pl (M.O.-G.); 3Department of Nephrology, Transplantology and Internal Diseases, Poznan University of Medical Sciences, 60-355 Poznań, Poland; zniemir@ump.edu.pl

**Keywords:** acute kidney injury, neutrophil gelatinase associated lipocalin, kidney injury molecule 1, interleukin 18

## Abstract

Cisplatin is still a widely used anticancer drug characterized by significant nephrotoxicity. Acute kidney injury (AKI), diagnosed based on the Kidney Disease: Improving Global Outcomes (KDIGO) criteria, has limitations, including a delayed increase in creatinine. We determined the usefulness of neutrophil gelatinase-associated lipocalin (NGAL), kidney injury molecule-1 (KIM-1), and interleukin-18 (IL-18) in diagnosing AKI according to the KDIGO criteria in patients treated with cisplatin. We recruited 21 subjects starting cisplatin-based chemotherapy (Cisplatin-based group) and 11 treated with carboplatin-based chemotherapy or 5-fluorouracil regimens (non-cisplatin-based group). Blood and urine samples were collected during four subsequent cycles of chemotherapy (68 and 38 cycles, respectively). AKI occurred in four patients in the cisplatin-based group (5.9% of 68 cisplatin-based chemotherapy cycles). Among them, three urinary markers were increased by over 100% in two cases, two in one case and one in another. A doubling of at least one investigated parameter was observed more frequently during cisplatin-based chemotherapy (80.3% vs. 52.8%; OR = 3.65, 95% CI: 1.49–8.90; *p* < 0.01). The doubling of at least one new urinary AKI marker was more common in patients receiving cisplatin and frequently was not associated with overt AKI. Thus, a subclinical kidney injury detected by these markers occurs more frequently than deterioration in kidney function stated with creatinine changes.

## 1. Introduction

Cisplatin is an anticancer drug introduced to therapy nearly 40 years ago. It is still widely used in treating solid tumors such as ovarian, testicular, head, and neck, as well as lung, and bladder cancers. Nephrotoxicity is becoming the most significant concern among its adverse drug reactions (ADRs), including neurotoxicity, ototoxicity, myelosuppression, nausea, and vomiting. Cisplatin is filtered throughout the glomeruli and then absorbed from primary urine by epithelial cells of proximal tubules via an organic cation transporter 2 (OCT2), reaching concentrations five times greater than in the serum [1]. The nephrotoxicity rate, at the beginning reported at the level of 30%, can be decreased using premedication with intravenous hydration and the use of 5-hydroxytryptamine receptor 3 (5-HT3) or neurokinin 1 (NK1) antagonists preventing cisplatin-induced nausea and vomiting resulting in dehydration. In addition, magnesium supplementation, decreasing organic cation transporter-2 (OCT2) expression in the proximal tubule epithelial cells, occurred helpfully [2,3]. Nevertheless, cisplatin nephrotoxicity is still a significant factor limiting its use in oncologic therapy. 

Acute kidney injury (AKI), diagnosed according to the Kidney Disease: Improving Global Outcomes (KDIGO) criteria, includes changes in serum creatinine concentration or urine output over time [4]. Despite its wide use, this method has several limitations. First, serum creatinine concentration is affected by sex, muscle mass, diet, and hydration. Secondly, an increase in serum creatinine (sCr) appears 48–72 h after the induction of AKI, which delays the diagnosis. The urine output criterion is not helpful in patients who are not catheterized and omits patients with non-oliguric AKI.

A new kidney injury marker should detect AKI earlier than serial measurements of sCr. Among the studied markers, neutrophil gelatinase-associated lipocalin (NGAL), kidney injury molecule-1 (KIM-1), and interleukin-18 (IL-18) are the most studied and best characterized potential candidates. 

NGAL, also known as lipocalin two or siderocalin, is a 25-kD protein primarily detected in neutrophil granules, expressed in several tissues, such as bone marrow, trachea, lungs, stomach, small and large intestine, pancreas, and kidneys [5,6]. This protein can be detected in serum and urine, where it occurs in monomeric form (main form in urine), and homodimer or heterodimers, found in serum, where they bind matrix metalloproteinase 9 (gelatinase) [7,8]. In normal conditions, urinary NGAL has two resources: the first, synthesized and secreted by tubular epithelial cells of the proximal and distal segment, and the second, originating from the circulation and filtrated by the glomerulus that is not reabsorbed in the proximal tubule [9,10]. In kidney injury, two mechanisms lead to increased urinary NGAL excretion. The first is the overexpression of NGAL mRNA in the thick ascending limb of Henle’s loop and the collecting ducts [9,11]. The second one is tubular injury, leading to impaired reabsorption from primary urine [12].

KIM-1 is a type 1 transmembrane glycoprotein containing immunoglobulin-like and mucin domains [13]. In normal conditions, its expression in the kidney is low but increases in ischemic or toxic kidney injuries [14]. It acts as a phosphatidylserine receptor mediating phagocytosis of apoptotic bodies and oxidized lipids, clearing tubular lumen from cellular debris [15,16]. Activation of phagocytosis in epithelial cells reduces infiltration of macrophages and secretion of proinflammatory cytokines, limiting kidney injury. KIM-1 can be cleaved from the epithelial cell surface by matrix metalloproteinase type 1 and 3, thus allowing its detection in urine [17,18]. Increased shedding of KIM-1 results in decreased endocytosis of apoptotic bodies [19] and stimulates macrophage infiltration. The latter secrete the proinflammatory cytokines [19], worsening kidney injury outcomes.

IL-18 is a proinflammatory cytokine produced by hemopoietic and non-hemopoietic lines, i.e., macrophages, osteoblasts, intestinal epithelial cells, and Kupffer cells [20]. It is produced as a 24-kD inactive precursor, pro-IL-18, cleaved to a 19-kD active form by the IL-1β-converting enzyme (also named caspase-1) [21,22]. In clinical trials on patients after the cardio surgery procedure, measurements of urine IL-18 enable the detection of the development of AKI earlier than changes in sCr concentration [23,24]. In kidney transplant patients, IL-18 identifies cases with proximal tubular necrosis, a severe form of AKI [25]. Moreover, in animal models of kidney injury, inhibition of IL-18 by exogenous IL-18-binding protein or using IL-18 knockout animals has been shown to protect the kidneys from apoptosis reduction and decreased expression of proinflammatory cytokines [26,27,28,29]. These findings emphasize the significant role of IL-18 in the development and course of AKI.

Data utilizing the so-called structural markers of AKI in the early detection of kidney injury induced by cisplatin-based chemotherapy (CTH) are scarce and contradictory [30,31,32,33,34,35,36,37,38]. The lack of a single marker for early detection of AKI prompts the search for a panel of tests consisting of several markers. Therefore, this study aimed to evaluate the new kidney damage markers NGAL, KIM-1, and IL-18 in patients receiving nephrotoxic cisplatin-based CTH.

## 2. Results

### 2.1. Characteristics of Study Groups

Twenty-one patients receiving cisplatin-based CTH underwent observation during 68 cycles (11 patients for four cycles, five patients for three cycles, four patients for two cycles, and one patient for one cycle), and 11 patients receiving treatment other than cisplatin-based CTH during 38 cycles (7 patients for four cycles, three patients for three cycles and one patient for one cycle). Due to incomplete data, two cycles from each group were excluded from further analysis, leaving both groups comparable concerning basic features and factors that can affect kidney function. Table 1 presents the main characteristics of the study groups. The occurrence of diseases predisposing patients to the development of chronic kidney disease did not statistically differ between study groups. 

### 2.2. Episodes of Acute Kidney Injury 

During the study, AKI occurred in four patients (in all cases, it was the first stage of AKI according to the KDIGO criteria). All of them were on cisplatin-based therapy, and in each case, AKI occurred only once. Two patients were diagnosed during the first cycle, following two patients during the third cycle of CTH. These patients accounted for 5.9% (95% CI: 2.3%–14.2%) of all 68 observational episodes. Among them, three markers were increased in two cases, two in one case, and one in another (Table 2).

### 2.3. Marker Concentration Changes during the Cycles

In all groups, one, two, and three increased markers were noted in 38 (37.2%), 24 (23.5%), and 10 (9.8%) subjects, respectively. None of the investigated markers (according to the working definition) increased at 24 h or 48 h after all patients initiated CTH during 30 CTH episodes. Only 13 (19.7%) were from the cisplatin-based CTH group. Conversely, episodes with the doubling of at least one or more investigated parameters (72 in the whole group) were observed more frequently during cisplatin-based CTH (80.3% vs. 52.8%; OR = 3.65, 95% CI: 1.49–8.90; *p* < 0.01). We noted a statistically significant difference in the occurrence of several positive markers between the cisplatin and non-cisplatin group (*p* < 0.01)—Figure 1. In the non-cisplatin CTH group, no subject had increased all three markers.

In the cisplatin group, a doubling of at least one parameter appeared during 38 episodes, two during 24 episodes, and all during 10 episodes. Notwithstanding the changes in serum creatinine, after 48 h and 72 h, they were similar regardless of the number of increased markers. The most significant increases (median 204%) among the investigated markers occurred for KIM-1 after 48 h.

Figure 2, Figure 3 and Figure 4 present the excretion of urinary markers standardized to creatinine levels. For NGAL, KIM-1, and IL-18, analysis of variance showed no significant differences between groups (cisplatin vs. non-cisplatin CTH group; *p* = 0.97, *p* = 0.20, and *p* = 0.56, respectively), but did show the significant influence of time (initial, 24 h and 48 h; *p* < 0.001) and interaction (*p* < 0.001). In the case of the group with three positive markers, NGAL values significantly increased between each time point (*p*-value at least < 0.05). We noted no other differences. For KIM-1, in the group with positive markers (1–3), the highest values were occurred after 48 h, with these values being significantly higher than the initial ones (*p* < 0.001), and after 24 h (*p* < 0.001). For IL-18, independent of the number of positive markers, the values after 24 h and 48 h were significantly higher than the initial estimates (*p*-value at least < 0.05). In the group with three positive markers, the values after 48 h were also higher than those observed after 24 h (*p* < 0.01).

Table 3 shows relative changes (Δ) from the initial values of all analyzed markers and serum creatinine. For NGAL, relative changes after 24 h and 48 h were significantly higher (*p* < 0.01) in the group with three positive markers compared to the group without positive ones. Moreover, Δ after 48 h in the group with three positive markers was significantly higher (*p* < 0.001) than in the group with only one positive marker. In the case of KIM-1, Δ values after 48 h in the group with no positive markers were significantly lower (negative) than in all other groups (*p*-value at least 0.05). Similarly, in the case of IL-18, Δ values after 24 h and 48 h in the group with no positive markers were significantly lower (negative) than in all other groups (*p*-value at least 0.01). No differences between groups were observed for relative changes in sCr levels.

In the group of patients with elevated levels of any tested marker, KIM-1 was most frequently elevated (64.2%). Table 4 shows a summary of the combinations of elevated markers.

## 3. Discussion

According to the literature, the incidence of AKI during cisplatin CTH ranges from 1.7% to 90% [30,31,32,33,34,35,36,37,38]. Such a large variability in published findings is the result of differential cisplatin dosing (25–120 mg/m^2^), the use of arbitrarily adopted diagnostic criteria of AKI (an increase in serum creatinine >25% from the baseline, an increase in blood urea nitrogen >20% from the baseline, a decrease in eGFR >20% from the baseline [30,32,35]), and the use of different measuring time points (e.g., only at one-time point [31,34,38]). Moreover, numerous factors may affect the occurrence of AKI in addition to high cisplatin dosage (>50 mg/m^2^), such as the frequency of administration, including cumulative dose, older age, comorbidities (hypertension, chronic kidney disease, diabetes, cardiovascular diseases), malnutrition, and low functional status [39,40,41]. 

In our study, AKI was diagnosed according to the commonly accepted KDIGO criteria, with material collected at three time points after the end of the cytostatic infusion. In addition, since the study comprised four cycles of CTH, the AKI cases were related to the number of consecutive episodes rather than the baseline number of patients, which would result in an overestimation of the occurrence of AKI (5.9% vs. 19.0%). The cisplatin dose in the study population ranged from 25 to 100 mg/m^2^, which depended on the variety of treated neoplasms and CTH regimens used. In studies including more homogenous groups of patients with head and neck tumors receiving the same CTH regimen with a high cisplatin dose of 100 mg/m^2^, the reported incidence of AKI varied between 34.1% and 68.5% [40,41,42]. In both of these studies, follow-up comprised more than one cycle (1–3), but the incidence of AKI was reported based on the number of patients enrolled. Considering the number of follow-up episodes, the incidence of AKI diagnosis would vary between 18.0 and 34.2% (own estimations). The incidence of AKI in patients treated with cisplatin may also depend on numerous factors related to the patient and the therapy itself. Therefore, when analyzing the incidence of AKI in this population, the results should always be interpreted in a broader context, e.g., cisplatin dosage, number of administered CTH cycles, comorbidities and comedications that could affect kidney function.

Initially, studies on NGAL, KIM-1, and IL-18 in patients undergoing cardiac surgery showed their advantage over serum creatinine measurements in the early diagnosis of AKI [23,24,43,44,45]. Another step in assessing the usefulness of the new AKI markers was their evaluation in non-ischemic kidney damage [23,31,34,35,37]. Nephrotoxicity is a side effect of numerous drugs, and one of the most significant risks is associated with the use of cisplatin [46,47]. The results of subsequent studies in oncologic patients treated with cisplatin were inconclusive. On the one hand, some studies showed that new markers, such as NGAL, KIM-1, and IL-18, could detect AKI [31,33,34,35,36,37,48,49,50,51]. However, the results of other studies were contradictory [30,32,38,50,52]. The observed discrepancies might be due to different locations of ischemic and toxic damage in the kidney. During ischemic kidney injury, the S3 segment of the proximal tubule and the thick ascending limb of Henle’s loop, both located in the outer medulla, are damaged due to their increased sensitivity to lack of oxygen [53,54,55]. In the case of toxic kidney injury during cisplatin treatment, the damage features are located only within the S3 segment of the proximal tubule [56,57]. A different damage location may lead to a different pattern of new kidney injury markers secretion. Moreover, new markers of kidney injury are proteins expressed *de novo* in response to kidney damage. Each of these proteins activates different, independent mechanisms participating in AKI development. What should be emphasized is that due to the complex process leading to the development of AKI, not all new markers have to be expressed in each case. The above findings may explain differences in research on new AKI markers, where some results confirmed their ability to detect AKI, and some did not. Therefore, it seems reasonable to propose a panel of tests consisting of several markers to find a new tool for the early diagnosis of AKI instead of only one. Additionally, AKI, defined by the KDIGO criteria, is a condition in which kidney damage is significant enough to increase serum creatinine. More than 50% of the nephrons of normally functioning kidneys must be damaged to increase serum creatinine [58,59]. If less than 50% of the nephrons are damaged, there may be an increase in the concentration of new AKI markers without an increase in serum creatinine concentration. The diagnosis of kidney damage based on new markers is a broader concept than that based on changes in creatinine concentration meeting KDIGO criteria. This condition can be described as a subclinical acute kidney injury (sAKI). Notably, no well-established definition of structural kidney injury, or sAKI is based on the new AKI markers. Regardless, in our study, the doubling of one, two, or even three markers was not associated with an increase in serum creatinine. The administration of iodine contrast agents is also more often associated with an increase in structural markers (sAKI) than with the development of overt AKI [60,61,62]. In studies of intensive care unit patients, sAKI was associated with a higher mortality rate than in non-AKI patients and was even comparable to patients diagnosed with AKI [63,64]. Thus, sAKI is an important clinical condition affecting the course of treatment. Our results also point to possible future directions for AKI prediction methods, integrating structural kidney injury biomarkers and relevant clinical data in large multicenter trials. Introducing structural markers into clinical practice, allowing for the diagnosis of sAKI, would enable early implementation of nephroprotective treatment and eventual modification of nephrotoxic systemic therapy schedules in individual patients.

Some limitations of our study have to be addressed. Namely, a lower-than-expected number of patients with AKI should be acknowledged in our study group. This result is probably the effect of an extensive hydration procedure before cisplatin administration, which is utilized in our center. This study limitation may be overcome in a multi-center study. In addition, we cannot exclude the impact of patient’s co-medication and other medical procedures with related nephrotoxicity, e.g., contrast-enhanced computer tomography before CTH cycles, which may predispose to cisplatin nephrotoxicity.

In conclusion, in our study, the doubling of at least one new urinary AKI marker was more common in patients receiving potentially nephrotoxic CHT. However, it was not frequently associated with overt AKI occurrence according to the established criteria. Thus, the increase in AKI’s structural markers suggests that a subclinical kidney injury occurs more frequently than the deterioration in kidney function identified by creatinine changes.

## 4. Materials and Methods

### 4.1. Study Population

We recruited 21 consecutive adult patients hospitalized in the Department of Internal Diseases and Oncological Chemotherapy in Katowice between October 2015 and December 2017, starting cisplatin-based CTH (Cisplatin-based group) and 11 patients treated with carboplatin-based CTH or de Gramont (5-fluorouracil) regimens (non-cisplatin-based group). There were no additional inclusion criteria. Patients with a history of previous CTH and chronic kidney disease (CKD, defined as eGFR <60 mL/min/1.73 m^2^ for more than three months) were excluded. 

### 4.2. Study Protocol

The protocol of this non-interventional study assumed additional blood sampling and collection of morning urine samples during four subsequent CTH cycles (C1, C2, C3, C4) in the following time points: directly before CTH (D1), and 24 h, 48 h, and 72 h after the beginning of CTH (D2, D3, D4; respectively). sCr and serum NGAL (SNGAL) levels were assessed in the blood samples, whereas urinary creatinine (uCr), NGAL (uNGAL), KIM-1 (uKIM-1), and IL-18 (uIL-18) concentrations were measured in the urine samples. The blood and urine samples were centrifuged for 10 min at 1000 rpm. The serum and urinary supernatants were stored at −70 °C until laboratory tests were performed. 

### 4.3. Measurement of NGAL, KIM-1, and IL-18

We used commercially available enzyme-linked immunosorbent assay kits to measure concentrations of serum and urinary monomeric SNGAL and uNGAL (Human NGAL ELISA Kit, Catalog Number: KIT 036CE, BioPorto Diagnostic, Hellerup, Denmark), uKIM-1 (Human KIM-1 ELISA Test Kit, Catalog Number: H-RENA-E-001, BioAssay Works, Tyler CT, Ijamsville, MD, USA), and uIL-18 (Human IL-18 Platinum ELISA, Catalog Number: BMS267/2CE, Affymetrix, Santa Clara, CA, USA), with intra-assay and inter-assay coefficients of variability: 10.5% and 7.5% for NGAL, <10% and <10% for KIM-1, and 6.5% and 8.1% for IL-18, respectively. All measurements were performed in the laboratory of the Department of Pathophysiology, Medical Faculty in Katowice, Medical University of Silesia in Katowice. Quality control was assessed based on repeated measurements and is reported as average deviation (4.4% for NGAL; 7.1% for KIM-1, and 8.7% for IL-18).

### 4.4. Data Analysis

The concentrations of the examined markers in the urine were related to creatinine excretion (to avoid the influence of the hydration status). Based on the data variability, we arbitrarily defined a 100% increase in any index after 24 h or 48 h as significant and potentially related to subclinical kidney injury. We distinguished subgroups with one, two, or three increased markers.

Acute kidney injury (AKI) was assessed based on the KDIGO criteria for each CTH cycle with at least 0.3 mg/dL or a 50% increase in sCr at any time point between Day 1 and Day 4 [4]. 

### 4.5. Statistical Analysis

Statistical analysis was performed using Statistica 12.0 (TIBCO Software Inc., Palo Alto, CA, USA) or R CRAN (Wirtschaftsuniversität Wien Wien, TU Dortmund, U Oxford, U Auckland) software (v. 3.0.1). The χ^2^ test with Yates correction was used to compare nominal data between study groups. A linear mixed model with fixed effects was used to calculate differences in concentration changes in AKI markers between the study groups. The mixed model was calculated with the log likelihood maximization procedure, random patient-specific effects, and identity correlation structure. Due to the skewed distribution of data, logarithmic transformation was performed. In both study groups, differences in markers between each day of the CTH cycle and mean values of consecutive days of the cycle were compared between cisplatin vs. the control group and between the first days of the cycle vs. other days, respectively. Multiple comparisons were corrected with the Benjamini-Hochberg procedure. Statistics results were considered statistically significant when *p*-values were less than 0.05.

## Figures and Tables

**Figure 1 ijms-25-01074-f001:**
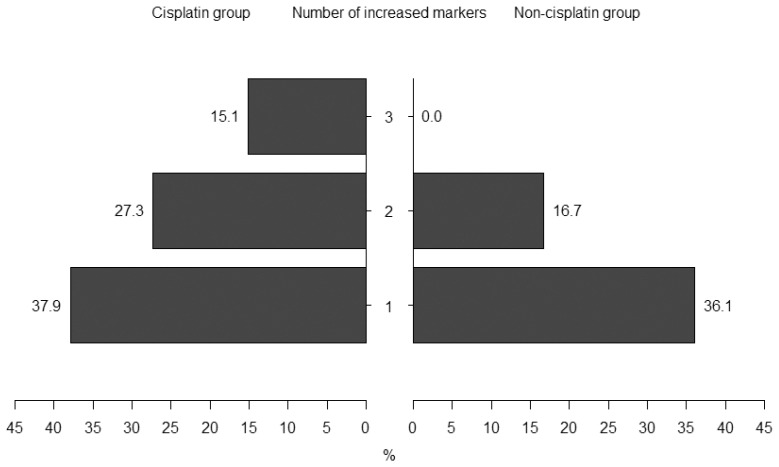
Percentages of patients depending on the number of positive urinary markers (at least 100% increase in KIM-1, IL-18, or NGAL to indexes after 24 h or 48 h) in the cisplatin and non-cisplatin groups.

**Figure 2 ijms-25-01074-f002:**
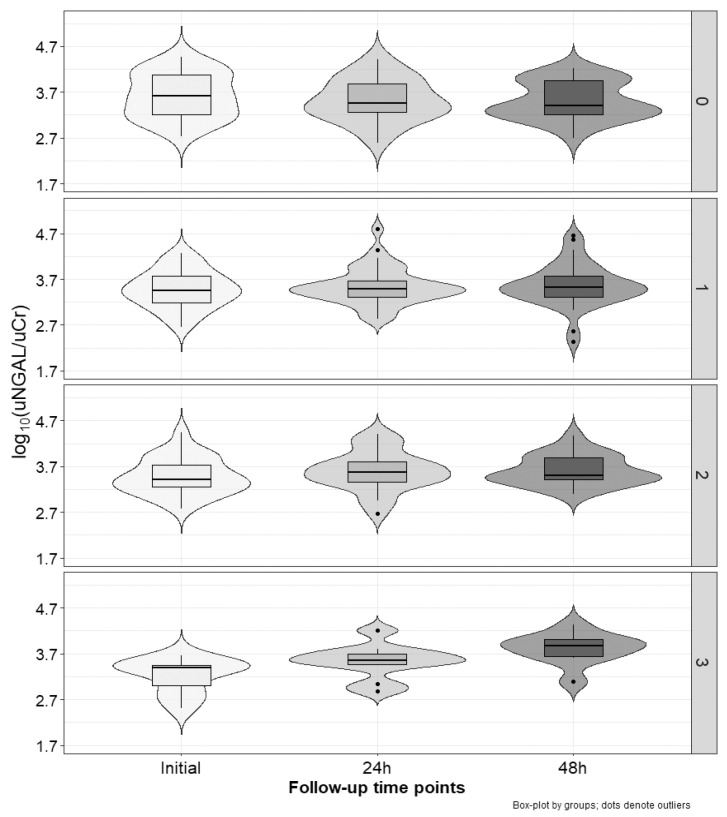
Changes in uNGAL/uCr index over time (x−axis) stratified by subgroups according to the number of increased markers (right y−axis): none (on the top), one, two, or three (on the bottom).

**Figure 3 ijms-25-01074-f003:**
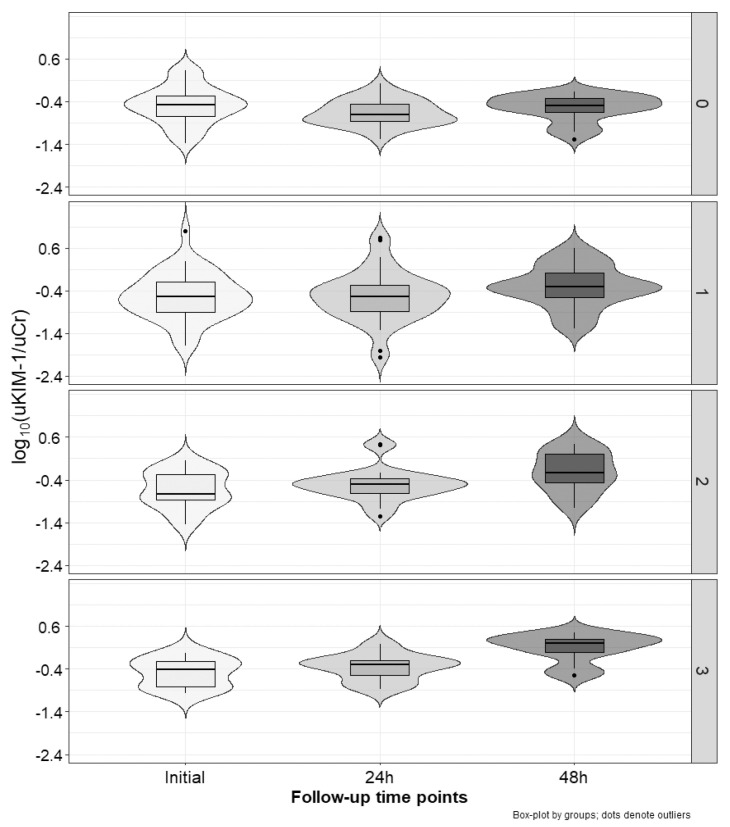
Changes in the uKIM-1/uCr index over time (x−axis) stratified by subgroups according to the number of increased markers (right y−axis): none (on the top), one, two, or three (on the bottom).

**Figure 4 ijms-25-01074-f004:**
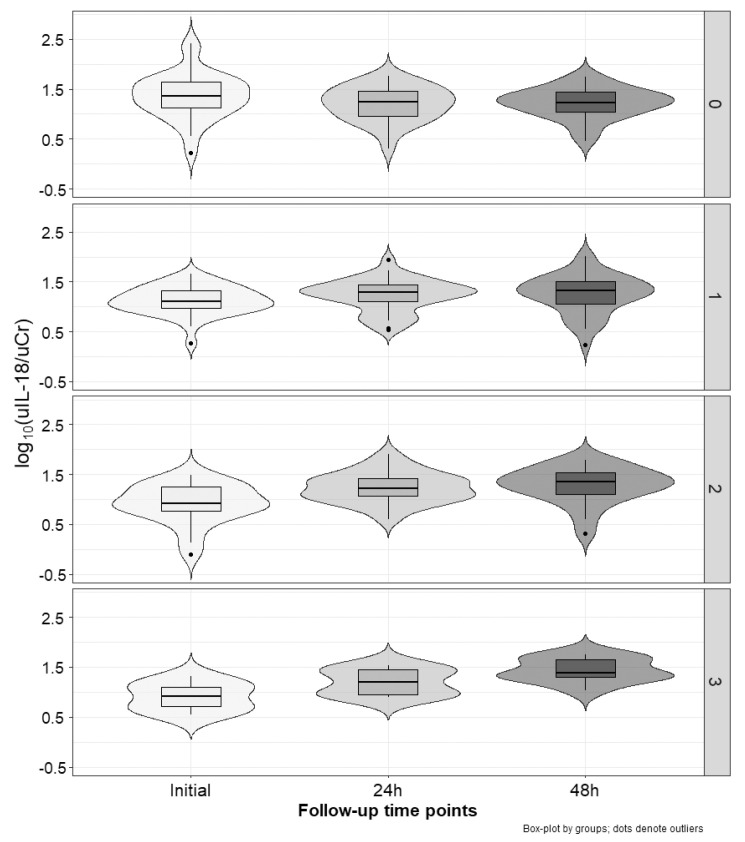
Changes in uIL-18/uCr index over time (x−axis) according to the number of increased markers (right y−axis): none (on the top), one, two, or three (on the bottom).

**Table 1 ijms-25-01074-t001:** Basic characteristics of cisplatin-based and non-cisplatin-based CTH groups.

	Cisplatin-Based CTH Group [*N* = *21*]	Non-Cisplatin Based CTH Group [*N* = *11*]	*p*
Men [N; (%)]	14 (66.7%)	5 (45.4%)	0.28
Age [yrs]	56 ± 8	58 ± 11	0.47
Body mass index [kg/m^2^]	25.3 ± 5.4	25.2 ± 4.7	0.96
Co-morbidity			
Hypertension [N; (%)]	8 (38.1%)	6 (54.6%)	0.46
Diabetes [N; (%)]	4 (19.0%)	4 (36.7%)	0.40
Coronary heart disease [N; (%)]	3 (14.3%)	3 (27.3%)	0.39
ACE-I [N; (%)]	5 (23.8%)	4 (36.4%)	0.68
Cisplatin dose per cycle [mg/m^2^]	69 ± 28.8	-	-
Cancer types [N; (%)]			
Head and neck	9 (42.9%)	-	-
Bile duct cancer	3 (14.3%)	-	-
Lung cancer	2 (9.5%)	1 (9.1%)	-
Gastric cancer	2 (9.5%)	-	-
Gallbladder cancer	2 (9.5%)	-	-
Urinary bladder cancer	1 (4.8%)	-	-
Ovary cancer	-	5 (45.4%)	-
Colon cancer	-	2 (18.2%)	-
Rectal cancer	-	1 (9.1%)	-
Pancreas cancer	-	1 (9.1%)	-
Cancer of unknown origin	2 (9.5%)	1 (9.1%)	-
Initial serum creatinine in C1 [mg/dL]	0.65 (0.55–0.91)	0.57 (0.42–0.91)	0.40
Initial uNGAL/uCr in C1 [ng/μmol]	2.85 (1.80–4.79)	3.42 (1.60–4.25)	0.78
Initial uKIM-1/uCr in C1 [ng/μmol]	0.22 (0.11–0.49)	0.20 (0.11–0.34)	0.72
Initial uIL-18/uCr in C1 [ng/μmol]	15.5 (8.2–20.5)	17.1 (13.0–29.4)	0.45
Cycle episodes [N]	68	36	-

**Table 2 ijms-25-01074-t002:** Changes in serum creatinine and urinary markers among four selected patients, in either the first (C1) or third (C3) cycle, with diagnosed AKI.

Patient Number (Cycle) /Measured Marker			Day 1	Day 2	Day 3	Day 4
015 (C3)	sCr	[mg/dL]	0.75	0.52	0.72	1.14
	uNGAL/uCr	[ng/μmol]	2.2	2.4	3.2	2.3
	uKIM-1/uCr	[ng/μmol]	0.35	0.25	0.58	**0.74**
	uIL-18/uCr	[ng/μmol]	9.5	**38.3**	**51.2**	**41.3**
016 (C1)	sCr	[mg/dL]	0.65	0.48	0.67	0.95
	uNGAL/uCr	[ng/μmol]	8.1	5.9	12.5	**18.4**
	uKIM-1/uCr	[ng/μmol]	0.66	0.49	**2.76**	**3.05**
	uIL-18/uCr	[ng/μmol]	7.2	13.0	**60.2**	**32.8**
017 (C3)	sCr	[mg/dL]	0.44	0.44	0.69	0.65
	uNGAL/uCr	[ng/μmol]	2.7	4.2	**36.8**	**130.5**
	uKIM-1/uCr	[ng/μmol]	1.28	1.21	1.62	-
	uIL-18/uCr	[ng/μmol]	16.7	13.6	17.2	23.9
032 (C1)	sCr	[mg/dL]	0.91	1.03	1.24	1.58
	uNGAL/uCr	[ng/μmol]	2.8	2.7	**10.4**	**8.1**
	uKIM-1/uCr	[ng/μmol]	0.95	0.61	**1.95**	**1.50**
	uIL-18/uCr	[ng/μmol]	11.4	**34.5**	**54.9**	**137.1**

Underlined are serum creatinine (sCr) values meeting the KDIGO criteria for the diagnosis of AKI. In bold are the values of investigated markers meeting the working definition of significant increase.

**Table 3 ijms-25-01074-t003:** Relative changes (after 24 h and 48 h) in the urinary levels of the analyzed markers and the serum levels of creatinine in groups without and with positive markers.

	Relative Values	No Increased Marker ^1^ N [%]	One Increased Marker ^1^ N [%]	Two Increased Markers ^1^ N [%]	Three Increased Markers ^1^ N [%]	*p*
**uNGAL/uCr**	Δ0–24 h [%] ^2^	5.2 (−50.3–44.1)	28.6 (−16.9–63.3)	42.4 (−23–112.4)	69.8 (34.5–159)	<0.01
	Δ0–48 h [%] ^3^	1.6 (−47.2–29.3)	9.1 (−28–99.1)	46.9 (−8.8–137)	185.5 (129.9–279.6)	<0.001
**uKIM-1/uCr**	Δ0–24 h [%] ^2^	−27.2 (−49.8–27.5)	10.1 (−36.2–41)	19.5 (−26.7–122.4)	25.5 (−2.1–94.7)	<0.01
	Δ0–48 h [%] ^3^	−2.5 (−28.8–27.1)	71.8 (26.6–149.5)	152.3 (107.1–261.7)	186.7 (160.8–355.9)	<0.001
**uIL-18/uCr**	Δ0–24 h [%] ^2^	−14.1 (−43.8–20.8)	33.4 (−13.6–77.5)	80.8 (30.1–158.5)	115.1 (30.8–199.2)	<0.001
	Δ0–48 h [%] ^3^	−9.3 (−36.5–15.7)	38.3 (−12.4–111.1)	136.9 (38.6–243.2)	261 (122.6–471.6)	<0.001
**sCr [mg/dL]**	Δ0–24 h [%] ^4^	−0.5 (−8.8–4)	1.6 (−10.9–6.1)	−2.6 (−7.1–5.2)	1.2 (−11.4–16.7)	0.89
	Δ0–48 h [%] ^5^	−3 (−11.3–3.1)	1.1 (−8.8–6.9)	−5.2 (−11.8–8.4)	−2.9 (−11.4–16.7)	0.74

^1^ Increase is defined as at least a 100% increase in the investigated markers between Day 0 and Day 1 or Day 2 of CTH. ^2^ The difference between mean values of the measured marker on Day 1 and Day 0. ^3^ The difference between mean values of the measured marker on Day 2 and Day 0. ^4^ The difference between mean serum creatinine values on Day 2 and Day 0. ^5^ The difference between mean serum creatinine values on Day 3 and Day 0.

**Table 4 ijms-25-01074-t004:** A combination of cases with any increased (+) urine markers; N = 72.

KIM-1 *N* = *47* [64.2% +]	IL-18 *N* = *39* [54.1% +]	NGAL *N* = *30* [41.6% +]	N [%]
+	-	-	17 [16.7]
+	+	-	14 [13.7]
+	-	+	6 [5.9]
+	+	+	10 [9.8]
-	+	-	11 [10.8]
-	-	+	10 [9.8]
-	+	+	4 [3.9]

## Data Availability

The data presented in this study are available on request from the corresponding author.

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
