# Peer review of "The Value of Urinary NGAL, KIM-1, and IL-18 Measurements in the Early Detection of Kidney Injury in Oncologic Patients Treated with Cisplatin-Based Chemotherapy"

_ijms, 2024, doi:10.3390/ijms25021074_

Round 1

Reviewer 1 Report

Comments and Suggestions for Authors

In this investigation, the researchers explored the effectiveness of NGAL, KIM-1, and IL-18 in diagnosing acute kidney injury (AKI) in patients undergoing cisplatin-based chemotherapy. The study comprised 21 participants in the cisplatin-based group and 11 in the non-cisplatin-based group. Applying the KDIGO criteria for AKI, the study revealed that AKI occurred in 5.9% of cisplatin-based chemotherapy cycles. Notably, the three urinary markers (NGAL, KIM-1, and IL-18) exhibited significant increases in some instances, with a more frequent doubling of at least one parameter observed in cisplatin-based chemotherapy. Intriguingly, the doubling of new urinary AKI markers was prevalent in patients receiving cisplatin, even in the absence of clinically evident AKI. The results suggest that subclinical kidney injury may manifest more frequently than changes in kidney function indicated by creatinine levels during cisplatin-based chemotherapy. While the study is compelling, it is noteworthy that the author cited relatively old references; thus, the author should incorporate more recent literature.

Reviewer 2 Report

Comments and Suggestions for Authors

Your manuscript investigating the utility of novel urinary biomarkers for the detection of subclinical kidney injury in patients undergoing cisplatin-based chemotherapy provides valuable contributions to the field of nephrology. The study design is sound, and the focus on subclinical kidney injury is commendable, as it addresses an important gap in current clinical practice.

There are several strengths in your study, including the rigorous sample collection protocol and the use of well-justified biomarkers. However, for the manuscript to reach its full potential, I recommend clarifying the rationale behind the definition of a 100% increase as a significant change in biomarker levels and providing a more detailed explanation of the statistical models used, including any adjustments for multiple testing.

While the sample size is small, which may limit the generalizability of the findings, the study lays a good foundation for larger, multi-center trials that could validate and extend these results. Furthermore, a discussion of how these biomarkers might be integrated into clinical practice would enhance the manuscript's impact.

In conclusion, with additional detail on the methods and a more thorough discussion of the implications and limitations, this study could significantly influence the monitoring and management of kidney health in patients receiving chemotherapy.

 Comments:

  1. Study Design and Population:
    • The non-interventional design is appropriate for the study's objectives. However, the small sample size may limit the generalizability of the findings. Future studies should aim for larger cohorts to validate these results.
    • The exclusion of patients with prior chemotherapy and chronic kidney disease (CKD) is justified to isolate the effects of cisplatin on kidney function. However, the study might benefit from including a subgroup of patients with pre-existing conditions to understand the compound effects.
  2. Biomarker Selection and Measurement:
    • The choice of NGAL, KIM-1, and IL-18 as biomarkers is well-founded, as they are recognized for their potential in early AKI detection. Nonetheless, it would be beneficial if the manuscript discussed the rationale behind the selection of these specific markers over others.
    • The application of ELISA is standard, but the manuscript should address the potential for variability and error inherent in this method, especially given the reported coefficients of variability. It would be advisable to have a quality control process in place to confirm the accuracy of these measurements.
  3. Data Analysis and Definition of Significant Change:
    • The arbitrary definition of a 100% increase as significant could be more rigorously justified. The choice of this threshold should be based on clinical relevance and prior research findings.
    • The adjustment of urinary marker concentrations for creatinine excretion is an important methodological strength that accounts for hydration status.
  4. Statistical Analysis:
    • The statistical methods employed are appropriate for the data analysis. However, given the multiple comparisons made, the manuscript should discuss whether adjustments for multiple testing were considered to control for the false discovery rate.
    • The use of a linear mixed model is suitable for this type of longitudinal data. However, the manuscript should provide more detail on the model specifications, such as the random effects included and the covariance structure used.
  5. Results Interpretation and Clinical Relevance:
    • The finding that the doubling of urinary AKI markers is not always associated with clinical AKI as defined by KDIGO suggests these markers could detect subclinical nephrotoxicity. The manuscript would benefit from a discussion on how these findings could impact current clinical practice and guidelines.
    • The clinical implications of detecting subclinical kidney injury should be explored further, especially regarding long-term kidney health and the potential for preventive interventions.
  6. Limitations:
    • The study acknowledges the limitation of not finding differences in serum creatinine levels. It should also address other limitations, such as the potential biases introduced by the non-randomized study design and the reliance on a single center for data collection.
    • The potential impact of other nephrotoxic medications or interventions concurrent with chemotherapy on the biomarkers should be discussed.
  7. Conclusions and Future Directions:
    • The conclusions drawn are cautious and reflect the data presented. It would be useful to suggest specific future research directions, perhaps involving multi-center trials or the development of predictive models integrating these biomarkers.
    • The manuscript could discuss the potential for these biomarkers in personalizing chemotherapy regimens to minimize nephrotoxicity.
  8. Overall Quality:
    • The manuscript is well-organized and addresses an important clinical issue. To improve the quality further, the authors should ensure that all conclusions are backed by the data and that the study's limitations are transparently discussed.

In summary, while the study provides valuable insights into the use of novel urinary biomarkers for detecting subclinical kidney injury, there is room for improving the robustness of the methodology and the clarity with which the results are presented and interpreted.

Reviewer 3 Report

Comments and Suggestions for Authors

This is a well designed study and well written manuscript. It is addressing the common issue of new biomarkers identifying AKI and pointing out the potential new direction of future research. 

Introduction is well organized however a little redundant and not necessarily focused on the question and conclusion. intro says to explore new biomarkers in cisplatin treated patients, but conclusion is less likely unique in cisplatin treated patients, more a universal finding of subclinical renal injury.

Study design, material and methods section is well described. 

Results section however not as satisfied.  Tables and numbers not always clearly described, such as table 2 and figure 1. the role of figs 2-4 is unclear and the interpretation of results also not well explained, which need to improve. 

Discussions and conclusion section is well written but better be more focused on the study findings. the conclusion is solely from table 3, however there's no explanation on why it is determined there were subclinical injuries. 

Comments on the Quality of English Language

Good writing skills. 

Round 2

Reviewer 2 Report

Comments and Suggestions for Authors

I am satisfied with the author's revision. I have no other queries.